# ErBb Family Proteins in Cholangiocarcinoma and Clinical Implications

**DOI:** 10.3390/jcm9072255

**Published:** 2020-07-16

**Authors:** Wook Jin

**Affiliations:** Laboratory of Molecular Disease and Cell Regulation, Department of Biochemistry, School of Medicine, Gachon University, Incheon 406-840, Korea; jinwo@gachon.ac.kr

**Keywords:** Epidermal growth factor receptor (EGFR) family, cholangiocarcinoma, aggressiveness, mutation, chemoresistance, EGFR family-targeting inhibitors

## Abstract

The erythroblastic leukemia viral oncogene homolog (ErBb) family consists of the receptor tyrosine kinases (RTK) epidermal growth factor receptor (EGFR; also called ERBB1), ERBB2, ERBB3, and ERBB4. This family is closely associated with the progression of cholangiocarcinoma (CC) through the regulation of cellular networks, which are enhanced during tumorigenesis, metastasis, and chemoresistance. Additionally, the constitutive activation of cellular signaling by the overexpression and somatic mutation-mediated alterations conferred by the ErBb family on cholangiocarcinoma and other cancers enhances tumor aggressiveness and chemoresistance by contributing to the tumor microenvironment. This review summarizes the recent findings on the molecular functions of the ErBb family and their mutations during the progression of cholangiocarcinoma. It also discusses the developments and applications of various devising strategies for targeting the ErBb family through different inhibitors in various stages of clinical trials, which are essential for improving targeted clinical therapies.

## 1. Introduction

Biliary tract cancers are highly lethal malignant tumors that include cholangiocarcinoma (CC) and gallbladder cancer (GBC). CC is an aggressive, rare tumor with a high mortality rate at late diagnosis that arises from the intrahepatic or extrahepatic biliary tract and is the second most common hepatic malignancy [1]. The five-year survival rate of patients with intrahepatic CC (ICC) after surgical resection is lower than 12.5% [2,3]. CC is anatomically classified into two subtypes, by its location as intrahepatic CC (ICC) arising from the liver or extrahepatic CC (ECC) from the extrahepatic bile ducts [4]. The overall incidence of ICC relative to ECC in the US markedly increased from 0.44 to 1.18 cases per 100,000 between 1973 and 2012 [5], and the overall incidence ratio (APC) of ICC in the US significantly increased between 2001 and 2015 compared with that of ECC (0.9 vs. 0.44) [6].

Alteration of the erythroblastic leukemia viral oncogene homolog (ErBb) family of the receptor tyrosine kinases (RTK) epidermal growth factor receptor (EGFR; also called ERBB1), ERBB2, ERBB3, and ERBB4 by comprehensive genomic or proteomic profiling has been identified in various cancer patients. EGFR and ERBBs have been implicated in the progression of various types of cancer through regulation of cell proliferation, survival, migration, angiogenesis, tumorigenesis, and metastasis by binding with ligands including epidermal growth factor (EGF), heparin-binding EGF-like growth factor (HB-EGF), transforming growth factor-α (TGFα), and β-cellulin as high-affinity ligands, in addition to amphiregulin (AREG), epiregulin (EREG), or epigen (EPGN) as low-affinity ligands [7,8]. Although members of the ERBB family are believed to be involved in cancer progression and have been identified in solid tumors and targeted in certain therapies for cancer treatment, much of their contribution to the progression of CC remains unknown relative to that of other types of tumors. This review discusses recent advances in the understanding of the biological functions of the ERBB family in the progression of CC and the efficacy of targeting this family in anti-cancer therapeutics.

## 2. Correlation between Epidermal Growth Factor or its Receptor and Cholangiocarcinoma Pathogenesis

EGF and its receptor EGFR are highly upregulated in 59.5% (22/37), and 32.4 (12/37) of patients with ICC, and these expressions showed no correlation with metastases of ICC. [9]. Another report revealed that EGFR is highly expressed in 44.7% (17/38) of ICC patients, and its upregulation is closely correlated with poor differentiation, lymph node metastasis, and aberrant p53 expression [10]. Expression of EGFR and vascular endothelial growth factor (VEGF) were highly elevated in 27.4% (29/109) and 53.8 (57/106) of ICC patients, respectively, and 19.2% (25/130) and 59.2% (77/130) of ECC patients, respectively [11,12]. EGFR is significantly associated with stage, lymph node metastasis, lymphatic vessel invasion, and perineural invasion in ECC. High EGFR expression also leads to reduced overall survival (OS) and increased tumor recurrence [11,12]. Moreover, HB-EGF and amphiregulin as EGFR ligands are highly elevated in various types of cancer cells, including CC cells, and significantly enhances the survival of cancer cells by activation of EGFR and the extracellular-signal-regulated kinase (ERK) pathway [13].

Based on morphology, ICC is subclassified into three subtypes: mass-forming ICC (MF-ICC), intraductal growth ICC (IG-ICC), and periductal infiltrating ICC (PI-ICC). IG-ICC is correlated with the most favorable outcome [14,15], and the upregulation of EGFR has been detected in 100% (9/9) of poorly differentiated non-intraductal growth ICC subtypes, which is correlated with a poorer outcome [16].

## 3. Correlation between Other EGFR Family Members and Cholangiocarcinoma Pathogenesis

Other ERBB family members are also involved in the pathogenesis of CC. ERBB2, ERBB3, and ERBB4 were significantly upregulated in 52.6% (20/28), 86.9% (33/38), and 39.5% (15/38) of ICC patients, respectively. The upregulation of ERBB3 and ERBB4 was significantly associated with lymph node metastasis [10]. Furthermore, ERBB2 was upregulated in 73% (46/63), 95% (59/63), and 75% (47/63) of CC cases, respectively. Also, ERBB2, c-myc, and RAS were positive in 86% (12/14) of hepatocellular carcinoma (HCC) patients, and c-myc and Ras have a direct correlation with bile duct malignancy and morphogenesis [17].

ERBB2 was also upregulated in 27.8% (5/18) [18], 98.4% (120/122) [19] of ICC and 63.3% (7/11) of GBC [18] cases. Expression of ERBB2 and c-Met, as the receptor for a hepatocyte growth factor (HGF), were markedly increased in tissue with cholangiofibrosis and hepatic intestinal metaplasias formed in the livers of rats, relative to normal and hyperplastic intrahepatic biliary epithelia [20]. ERBB2 and c-Met were highly expressed in 55% (45/81), and 35% (28/81) of ICC patients, respectively. These proteins have been reported to have a close correlation with lymph node metastasis as ERBB2 and c-Met were positive in 76% (19/25) and 16% (4/25) of metastatic lymph nodes in patients with ICC, and high c-MET expression was significantly correlated with poor survival of these patients [21]. Moreover, ERBB2 was overexpressed in 29.1% (16/55) [22], 80% (60/75) [23], and 82.4% (28/34) [19] of patients with ECC, and ERBB2 is closely associated with tumor grade, higher rates of distant metastases, and poor survival of ECC patients with lymphatic invasion and perineural invasion [22,23]. Furthermore, chronic bile duct injury by exposure to chronic intermittent toxins such as CCl_4_ provoked the development of ICC in p53-deficient rats by upregulation of oxidative stress markers cyclooxygenase (COX)-2 and inducible nitric oxide synthase (iNOS) in addition to ERBB2 and c-MET, with downregulation of E-cadherin [24].

ERBB3 was markedly overexpressed in 39% (90/230) of ECC patients and predominantly localized in the cytoplasm. ERBB3 is upregulated in nodular and infiltrative tumors relative to the papillary tumor and has been significantly associated with reduced survival of patients with ECC [25]. Another study showed that ERBB3 and ERBB4 were upregulated in 12.3% (8/65) and 63.1% (41/65) of patients with ICC and 11.8% (13/110) and 56.4% (62/110) of those with ECC, respectively. ERBB3 expression was significantly associated with poorly differentiated ICC, and ERBB4 expression increased the ability of CC cells to migrates [26]. ERBB3 was markedly upregulated in 10% (1/10) of patients with GBC, 23% (6/23) of patients with ICC, and 17% (3/18) of those with ECC, respectively, and was mainly overexpressed in the cytoplasm [27].

## 4. The Functional Role of the EGFR Family in the Progression of Cholangiocarcinoma

c-Cbl-mediated endocytosis and degradation of EGFR are closely associated with the major negative regulatory mechanism of EGFR through the formation of c-Cbl/EGFR complex [28]. In CC cells, however, EGFR signaling is sustained because the association of EGFR with c-Cbl is prevented, precluding c-Cbl-mediated EGFR degradation [29]. The resulting activation of EGFR enhances the progression of CC through its downstream oncogenes and targets involved in the proliferation and tumorigenesis of cancer (Figure 1).

Tescalcin (TESC) regulates cytoplasmic pH via interacting with the NA^+^/H^+^ exchanger and promotes the progression of colorectal cancer (CRC) by activating the AKT/NF-κB pathway [30]; upregulation of TESC is also involved in the pathogenesis of CC. TESC is highly upregulated in patients with ICC and is associated with poor survival. TESC induced by the TGFα/EGFR/signal transducer and activator of transcription 3 (STAT3) signaling pathway promotes cell proliferation and tumor growth in vivo and in vitro by increasing the expression of forkhead box M1 (FOXM1) [31]. Protein tyrosine kinase 6 (PTK6) expression is significantly elevated in patients with distal and perihilar CC (compared with patients with ICC or well-differentiated tumors) and colocalized with EGFR and ERBB2. EGF-mediated PTK6 activation promotes the proliferation of CC cells through the formation of PTK6/EGFR and PTK6/ERBB2 complexes, which in turn activate Src associated in mitosis of 68 kDa (SAM68); high levels of both PTK6 and EGFR expression are closely correlated with the upregulation of Ki67 as a prognostic marker of cancer [32].

EGFR activation is primarily involved in the malignant progression and chemotherapeutic resistance of cancer through the epithelial-mesenchymal transition (EMT), enhancing the mesenchymal fate by activation of the mitogen-activated protein kinase/ERK kinase (MEK)/ERK and Janus kinase (JAK)2/STAT3 pathways [33,34]. E-cadherin is preferentially localized in the cytoplasm of patients with CC relative to localization at the plasma membrane of bile duct epithelial cells, and its localization is significantly correlated with EGFR expression in patients with CC, such that loss of EGFR activity leads to restoration of membrane expression of E-cadherin. EGF-mediated EGFR activation in CC cells induces EMT via reducing E-cadherin and increasing mesenchymal markers such as N-cadherin and α-smooth muscle actin (α-SMA) by the induction of EMT-transcription factors (EMT-TFs), including Slug and Zeb1. However, treatment with gefitinib abrogates the metastatic ability of CC cells by EGF/EGFR-induced EMT [35]. Additional mechanisms underlying EGFR-induced progression of CC involve the direct activation of c-Met, the tyrosine kinase receptor for HGF which contributes to the malignancies of cancer [36]. EGFR family members induce c-Met activation via an HGF-independent manner by direct interaction with c-Met [37,38]. The upregulation of c-Met is closely correlated with the overexpression of EGFR and reduced survival in patients with ICC or ECC [39].

The invasion and proliferative ability of CC cells depend on the level of ERBB2 expression and the progression of CC cells by ERBB2-mediated activation of the AKT/p70S6K signaling pathway [40]. Tyrphostin AG1517 and tyrphostin AG879 as inhibitors of EGFR and ERBB2, respectively, effectively suppress the growth of CC cells, and combination treatment has shown a synergistic effect on the inhibition of CC cell growth by inhibiting cyclin D1 and activating caspase-3 [41]. ERBB2-induced gene expression profiles as determined by microarray analysis included cytokeratin 20 (Krt20), Sry-related HMG box gene-17 (Sox17), amphiregulin (Areg), MUC1, and sphingosine kinase 1 (Sphk1) among the genes strongly correlated with tumor growth and metastases of CC cells [42].

## 5. Modulators of EGFR Family Activation in the Progression of Cholangiocarcinoma

### 5.1. The Role of Other Ligands and its Receptor as Positive Modulators

Other ligands and receptors also cooperate in the activation of EGFR and ERBBs via induction of EGFR expression to induce critical cellular pathways in the progression of CC (Figure 2). For example, 5-aza-2′-deoxycytidine (5-aza-CdR), a DNA methylation inhibitor, inhibits the cell growth and invasion of CC cells, but interleukin (IL)-6 abrogates 5-aza-CdR-mediated inhibition of cell growth by regulating the activity of DNA methyltransferases (DNMT). Genome-wide microarray analysis revealed that the expression of EGFR and mitogen-activated protein kinase (MAPK) kinase 2 is regulated by IL-6 and 5-aza-CdR. IL-6 treatment significantly upregulated the expression of EGFR by decreasing the methylation of the EGFR promoter by reducing its sensitivity against 5-aza-CdR [43].

Furthermore, neurotensin (NTS), a neurotransmitter in the central nervous system (CNS), and NTS receptors (NTSR’s: NTSR1 and NTSR2) have been implicated in other cancers [44] and CC [45]. NTS was significantly elevated in 85% (34/40) of these patients and was increased in metastatic lesions relative to perinormal tissues. High NTS expression was closely correlated with reduced progression-free survival (PFS) and OS of patients with CC, and loss of NTS led to suppression of CC cell invasion through inhibition of the EGFR/AKT signaling pathway by reducing EGFR expression [45].

Moreover, mucins (MUCs) are transmembrane glycoproteins with high glycosylation of prolines, threonines, and serines (PTS domains); mucins are also involved in malignancy through regulation of cell growth and survival [46]. In one study, the upregulation of MUC1 was identified in 90% of MF-ICC cases. It was closely correlated with invasive proliferation and poor outcomes, and MUC2 was significantly elevated in 67% of PI-ICC cases and 86% of patients with bile duct cystadenocarcinoma (BDCC) with a favorable outcome [47]. In another study, MUC1 and MUC4, which are also ligands for RTK ERBB2 [46], were markedly upregulated in 81% (22/27) and 37% (10/27) of patients with MF-ICC, respectively, who displayed poorer prognoses than patients with IG-ICC or PI-ICC. High MUC4 and ERBB2 expression led to significantly reduced survival of patients with ICC [48]. Overexpression of ERBB2 in CC cells of a rat model markedly induced MUC1 and COX-2 and activated the AKT and MAPK signaling pathways. ERBB2 expression significantly increased tumor formation in vivo by increasing bile duct obstruction and gross peritoneal metastases [49]. Another mucin, mucin 13 (MUC13), promotes the progression of various cancer types, including CC, through induction of ERBB2 expression and NF-κB activation [50,51,52]. MUC13 was markedly elevated in 93.8% of patients with ICC relative to the corresponding adjacent non-tumor tissues, and the upregulation of MUC13 was closely correlated with reduced OS and PFS of patients with CC. Elevated MUC13 induces the metastatic ability, tumor growth, and lung metastasis of CC cells in vivo and in vitro by reducing tissue inhibitors of metalloproteinases (TIMP)1 and inducing matrix metalloprotease (MMP)9 expression by MUC13/EGFR complex-mediated activation of EGFR/AKT and reducing the expression of the microRNA miR-212-3p as a tumor suppressor [53].

Several receptors lead to EGFR activation by directly interacting with EGFR (Figure 2). Insulin-like growth factors (IGFs: IGF1 and IGF2) and insulin act as ligands of insulin-like growth factor-1 receptor (IGF-1R). IGFs and insulin-induced IGF-1R activation contribute to the progression of cancer and are current chemotherapeutic targets [54]. The crosstalk between EGFR and other growth factor receptors markedly enhances the progression of cancer by the transactivation of EGFR through interaction with platelet-derived growth factor (PDGFR) and IGF-1R [55,56,57]. CC cells that are resistant to the EGFR inhibitor erlotinib acquire cancer stem cell (CSC) properties via activation of IGF2/IR/IGF-1R and its downstream signaling to induce EMT. However, suppression of IGF2/IR/IGF-1R inhibits the metastatic ability and tumor growth of erlotinib-resistant CC cells in vivo and in vitro, and linstinib-mediated IGF-1R inhibition also markedly suppressed the proliferation of human liver myofibroblasts (HLMF) through reduction of α-SMA [58].

Neuropilin-1 (NRP1) is a transmembrane receptor that promotes the progression of cancer by interacting with VEGF/VEGFR2 [59] and EGFR [60]. NRP-1 is highly expressed in CC tissue relative to corresponding normal biliary tissues, and markedly reduces miR-320 expression, which is a negative regulator of oncogenes. Furthermore, loss of NRP-1 induces cell cycle arrest at phase G1/S by suppressing cyclin E and cyclin-dependent kinase (CDK)2 and inducing p27, and it inhibits tumor metastasis and growth in vivo and in vitro by inhibiting activation of the VEGF/VEGFR, EGF/EGFR, and HGF/c-Met pathways [61]. The upregulation of L1 cell adhesion molecule (L1CAM) enhances the activation of EGFR by interacting with EGFR, ERBB2, and ERBB3 and then increases resistance to cisplatin and proliferation of ICC cells through induction of its downstream signaling pathways including ERK and AKT [62,63]. L1CAM is a transmembrane glycoprotein that has been reported as highly expressed in 40.5% (17/42) of patients with poorly differentiated ICC, and loss of L1CAM markedly reduces the tumor growth and metastatic ability of CC cells in vivo and in vitro by inhibiting focal adhesion kinase (FAK) and AKT activation [64]. Additionally, the prostaglandin (PGE) 2-coupled E prostanoid (EP) 1 receptor activates EGFR by forming a scaffolding complex and facilitating the binding of EGFR to c-Src. These complexes lead to the induction of cell proliferation and invasion of CC cells via subsequent ATK activation. Inhibition of PGE2 synthesis or loss of the EP1 receptor suppresses EGFR activation [65].

G protein-coupled receptor (GPCR) signaling systems lead to the transactivation of EGFR via cross-talk between GPCR and EGFR to promote CC tumorigenesis (Figure 2). Angiotensin-converting enzyme (ACE) promotes the conversion to the active ligand, angiotensin II (Ang II), from angiotensin I [66], and Ang II and its receptor AT1R as a GPCR contribute to the pathogenesis of CC. Serum ACE levels were drastically higher in patients with ECC relative to those of healthy individuals [67], and Ang II levels were 3.7-fold greater in patients with ICC relative to those with HCC or with normal livers. AT1R was significantly upregulated in 87.5% (14/16) of patients with ICC and lead to the proliferation of CC cells [68]. Treatment with Ang II markedly induces stromal cell-derived factor-1 (SDF-1) as a ligand of CXCR4, increases the proliferation of hepatic stellate cells (HSCs) by SDF-1/CXCR4 activation, and promotes HSC-mediated ICC progression through induction of EMT [69]. AT1R blockers (ARBs) have beneficial effects on tumor growth, vascularization, and metastasis by inhibiting AT1R-mediated transactivation of EGFR and induction of VEGF [66]. Telmisartan, an ARB, induces apoptosis of CC cells by reducing cell cycle-related proteins (G1 cyclin, cyclin D1, CDK4, and CDK6), which induces cell-cycle arrest and suppressing tumor growth of CC cells in vivo by inhibiting activation of EGFR and TIMP-1 [70].

### 5.2. The Role of Other Positive Modulators of EGFR

EGFR activation has been reported to be induced by multiple sources from tumor microenvironments, including tissue-resident myofibroblasts, bile acids (BA’s), secreted molecules, and cellular components (Figure 3).

Paracrine factors, including HB-EGF secreted by HLMFs and bile acids synthesized in the liver, contribute to the pathogenesis of CC. The role of EGFR has been identified in mechanistic cross-talk between myofibroblasts (MF) and CC cells. Increased HB-EGF secretion in HLMFs, which upregulate α-SMA, markedly increases tumor formation of CC cells and tumor invasion in vivo by EGFR activation-mediated upregulation of PECAM-1 and activation of STAT3/ERK. Additionally, EGFR activation-mediated TGF-β1 production by CC cells elevates α-SMA expression and induces HLMF activation, eventually enhancing more myofibroblastic morphology [71].

BA’s are hydroxylated steroids, including cholic acid (CA), deoxycholic acid (DCA), taurodeoxycholic acid, taurolithocholic acid (TLCA), taurocholic acid (TCA), glycocholic acid (GCA), glycodeoxycholic acid (GDCA), and lithocholic acid (LCA), which are synthesized in the liver from cholesterol, and 95% of BA’s are reabsorbed in the liver from the intestines. BAs promote the pathogenesis of gastrointestinal cancers, including CRC and HCC, by activating signaling pathways such as EGFR/Ras/ERK1/2 [72,73,74]. Treatment of BAs in CC cells significantly induces EGFR activation and subsequently increases COX-2 expression through activation of EGFR downstream targets, including MAPK and p38 signaling pathways [75]. Additionally, treatment with BAs enhances the survival of CC cells through blockage of Mcl-1 degradation as antiapoptotic Bcl-2 family protein by EGFR/Raf1 activation [76]. Among the EGFR ligands, the release of TGF-α by BA-mediated MMP activity induces the growth of CC cells via EGFR activation [77]. Taurolithocholic acid (TLCA) as a metabolite of BA taurine increases cell growth through induction of the EGFR/ERK1/2 signaling pathway by activation of muscarinic acetylcholine receptor M3 (M3 mAChR)-mediated EGFR [78]. Another conjugated BA, taurocholate (TCA), induces COX-2 expression through activation of AKT/ERK1/2/NFκB signaling by activating sphingosine 1-phosphate receptor 2 (S1PR2), which promotes the invasion and cell growth of CC cells. Also, TCA-mediated S1PR2 activation indirectly induces GPCR-mediated activation of EGFR by upregulation of MMP2 and MMP9 and subsequently increases the cell growth of CC cells through induction of COX-2 expression via activation of the AKT/ERK1/2 signaling pathway [79]. Blocking EGFR degradation induces the proliferation of CC cells through prolonged EGFR activation-mediated COX-2/prostaglandin (PGE2) signaling relative to other hepatoma cells, and PGE2 as a downstream product of COX-2 prevents EGFR inhibitor-mediated growth suppression in CC cells [80]. EGFR and COX-2 are significantly increased in CC tissues relative to the non-neoplastic bile duct epithelium, and upregulation of COX-2 increases PGE2 synthesis and subsequently markedly transactivates EGFR [65]. EGFR, COX-2, AKT, and p-MAPK were significantly upregulated in patients with CC (62.5% (15/23), 100% (23/23), 95.8% (23/24), and 87% (20/23), respectively), and these expression patterns exhibit a significant correlation between EGFR and AKT expression in these patients [81]. Additionally, EP1 receptor-induced intracellular calcium concentration induces EGFR activation and subsequently leads to PGE2-mediated ERK activation [82]. Moreover, PGE2-mediated activation of the EP1 receptor induces activation of the Ca^2+^/EGFR/Erk pathway and then increases MMP2, leading to cell proliferation of CC cells by activation of cyclic AMP response element-binding protein (CREB), transcription factor [83]. Moreover, PGE2-mediated activation of the EP3 isoform 4 receptor also enhances the progression of CC cells through the upregulation of β-catenin via the Src/EGFR/PI3K/AKT/GSK-3β pathway [84]. Treatment with CA and DCA as BAs induces the migration of CC cells, and CA- and DCA-mediated protein expression profiles identify CCD25 and is significantly upregulated in patients with CC. Also, CCD25 has been associated with the son of sevenless homolog 1 (SOS1) and growth factor receptor-bound protein 2 (GRB2), which are involved in EGFR signaling [85].

Engagement of positive modulators of EGFR enhances the progression of CC via EGFR-induced EMT programming. Fucosylation, which is the most common modification of glycoproteins and glycolipids, induces the biosynthesis of glycan branches of proteins via the addition of fucose to the reducing end of N- and O-linked glycan structures in glycoproteins and glycolipids by fucosyltransferases (FUTs). Fucosylated glycan is closely associated with tumor progression [86,87]. High levels of fucosylation (the terminal α1, 2-fucose containing glycan (TFG)) are identified in CC patients and significantly associated with reduced survival. Loss of FUT1 reduces the metastatic ability of CC cells by inhibiting EMT through suppression of EGFR-mediated activation of the AKT/ERK pathway [88]. The trefoil factor family (TFF’s) also contributes to the restitution and repair of the gastric and intestinal epithelium against mucosal injury [89] and promotes tumor pathogenesis in various cancer types through activation of several pathways and induction of EMT [90,91]. The expression of TFF1, TFF2, and TFF3 was markedly upregulated in 50.9% (56/110), 25.5% (28/110), and 49.1% (54/110) in CC tissues relative to normal tissues, respectively. TFF2 induces EGFR activation and then promotes cell proliferation via EGFR-mediated MAPK activation [92]. Moreover, SOX4 expression correlated with the clinicopathology of CC. SOX4, which has been associated with poor survival of patients with CC, was upregulated in 29.3% (17/58) of ICC and 29.8% (28/94) of ECC cases, respectively, and SOX4 expression was closely associated with upregulation of EGFR in both ECC and ICC groups, and patients with CC upregulating both SOX4 and EGFR showed the worst survival. Loss of SOX4 in CC cells markedly reduced the migration and EMT of CC cells by suppressing EGFR expression [93].

Laminin subunit gamma 2 (LAMC2) as a positive modulator leads to the activation of EGFR through direct interaction. LAMC2 enhances the progression of CC by inducing the EGFR signaling pathway. LAMC2 was highly expressed in 59.5% (72/121) of patients with CC and was closely related to the invasion and tumor-node-metastasis of these patients. LAMC2 promotes metastatic ability and angiogenesis of CC cells through induction of VEGFR expression and EGFR activation-mediated EMT induction by the formation of an EGFR/LAMC2 complex [94,95].

### 5.3. The role of Negative Modulators in EGFR Family Activation

Merlin, the neurofibromatosis type 2 (Nf2) tumor suppressor, inhibits activation of EGFR and suppresses the internalization of EGFR by NHE-RH1-mediated Merlin-EGFR association, which subsequently leads to the proliferation of CC cells by preventing the formation of EGFR complexes with its targets [96]. NF^−/−^ mice developed CC and HCC through the over proliferation of Nf2^−/−^ liver progenitors by aberrant activation of EGFR and its signaling targets such as STAT3 and AKT [97].

The scaffolding protein EBP50 also is involved in the pathogenesis of cancer by interacting with EGFR and PDGFRβ with its PDZ domains [98]. EBP50 is delocalized to the cytoplasm in CC tissues relative to the normal biliary epithelium. Loss of EBP50 enhances cell motility and migration through induction CC cell EMT by increasing activation of EGFR and its signaling pathways such as ERK1/2 and STAT3 [99].

## 6. EGFR Family Mutation and its Functional Roles During Cancer Progression

Identification of mutations in various types of cancers by next-generation sequencing technology has demonstrated that somatic mutations as activating mutations are significantly correlated with metastases and poor outcomes in targeted therapy via constitutive activation of downstream signaling pathways, eventually leading to new strategies for mutation-driven drug resistance [100,101].

EGFR mutations have been identified in CC, but the functional impact of these mutations remains largely unknown in the pathogenesis of CC (Figure 4 and Table 1). The overall mutation rate of EGFR was identified in 13.6% (3/322) of patients with CC. All identified EGFR mutations were in-frame deletions in exon 19 (K745_E749 del) of patients with ICC and exon 19 (E746_A750 del) of one patient with ECC. These mutations are closely correlated with the poor survival of these patients [102]. It was also shown that 15% (6/40) of patients with CC had point mutations in the tyrosine kinase domain of the EGFR of patients with ICC (K575R, E872K, T790M), ECC (C775Y, G882S, V843I, L858R), and GBC (A864T). Additionally, 90% (36/40) of CC patients showed a silent mutation at codon 787 in exon 20, and 83.3% of patients with EGFR mutations showed increased MAPK and AKT phosphorylation [103]. Another study demonstrated that 5% (1/20) of patients with CC had a mutation (E804K) in the tyrosine kinase domain located in exon 20 of EGFR, but its function is unknown [104].

Another EGFR (G719S) mutation was identified in 1.5% (1/94) of patients with ICC, and this mutation showed a high degree of EGFR amplification. The patient that harbored this mutation had pulmonary metastasis after undergoing curative surgical resection [105]. Moreover, EGFR mutations (E709K, L747–P753 delins, V786M) were identified in 7.4% (6/81) of patients with ICC, and a higher EGFR mutation rate (5/38, 13.2%) was detected in patients with ICC who had chronic advanced liver disease relative to those with normal livers (1/43, 2.3%) [106]. Another report exhibits that mutations of EGFR (G719X, S768I, and L861Q) were identified in 21% (17/81) of patients with CC [107]. Additionally, 9.5% (13/137) of biliary tract cancer (BTC) patients including those with CC have mutations in exon 20 and exon 21 in patients with ICC (T783I, S784F, D837N), ECC (D800G, C818R, V819M, Q820R, D837N, V851I, G873E, G874D), or GBC (A837N, T785I) [108], which correspond with the kinase domain of EGFR [108]. Also, 10.5% (6/57) and 12.3% (7/57) of BTC patients have an identified mutation in the extracellular domain (ECD) of EGFR (L443Q, S464P, K467stop, N468D, G482E, G482R, L469S) as new mutations and in the tyrosine kinase domain (TKD) of EGFR (L707S, V786M, L788H, G810S, G824S, D837N T854I, D855N), respectively; 50%, 16.7%, and 33.3% of mutated BTC were detected in patients with ICC, ECC, and GBC, respectively. Also, patients with a mutation of the ECD of EGFR revealed a worse OS, and those with maturation at the TKD of EGFR had shorter PFS and OS [109]. Moreover, new mutations of EGFR were identified in patients with BTCs, including CC (E114K, Y1069C, I425L, C818F, G203R, R669Qfs*36, V524Sfs*44) [110,111].

ERBB2 mutations were identified in 25% (5/20) of patients with ECC, wherein ERBB2 was mutated in the kinase domain (V777L) and extracellular domain (S310F) [113]. Also, ERBB2 activating mutations (S310F/Y, G292R, T862A, D769H, L869R, V842I, R678Q, G776V, S653C, R897W, and G660D) were detected in 2% (9/459) of patients with CC. Although mutations of ERBB4 have been identified in patients with CC (S79Y, R106C, D376Y, C580*, K682N, F682L, E835D, R847C, R938C, Y950H, D960G, R992H, Y1066H, P1092S, Q1126K, Q1270K), its function remains unknown [112]. Moreover, new mutations of ERBB2 have been identified in patients with BTC, including CC (L755P/S, E265K, L994V, L1098M) (TCGA dataset) [111]. ERBB3 mutations (G284R, V104M, A232V, E928G, G284R, V104L, D297Y, T355I) were detected in patients with BTC including CC, as activating mutants. Other mutants of EBBB3 (V1035D, G508R, G582W, A1252S, D581N, R444Q, V586M, G994D, N222Tfs*47, E230Dfs*39, D73Tfs*11, D296Ifs*16, D297Ifs*16) and ERBB4 (R103C, I68N, L432M, S602C, P1158H, T475A, S1286Lfs*5, F356Sfs*2) were identified in patients with BTC, including CC, but their function is unknown (TCGA dataset) [110,111].

## 7. EGFR Inhibitors and Their Efficacy in Cholangiocarcinoma

EGFR inhibitors and their efficacy in cholangiocarcinoma summarized in Table 2.

**Imatinib**. Imatinib is an inhibitor of platelet-derived growth factor receptor PDGFR, c-Abl, or c-Kit, but CC cells do not detect these proteins. PDGFRβ leads to transactivation of EGFR via the formation of a PDGFRβ/EGFR complex [56], and imatinib treatment inhibits PDGFRβ-mediated cell proliferation and tumor growth in CC cells [114]. Treatment with imatinib suppresses EGFR/FAK/AKT activation and reduces the survival of CC cells by reducing Mcl-1 expression [115]. Imatinib, in combination with 5FU/Leucovorin, has also been tested in clinical trials, but results have not yet been published (ClinicalTrials.gov Identifier: NCT01153750).

**Gefitinib**. The FDA approved gefitinib is an EGFR inhibitor for the treatment of non-small cell lung cancer (NSCLC) with EGFR mutations, which has exon 19 deletions or exon 21 (L858R) [136]. CC cells showed resistance to gefitinib or CI-1040, an inhibitor of MEK1/2, alone. Gefitinib effectively inhibits EGFR but not ERK1/2 in xenografted tissues, and CI-1040 treatment showed a slight inhibition of ERK1/2 activation. However, combination therapy with gefitinib and CI-1040 significantly suppressed ~60% of tumor growth of CC cells by significant inhibition of both EGFR/AKT and ERK1/2 [116]. At high concentrations (≥1 μM), gefitinib markedly suppresses TGFα-induced growth of CC cells. Gemcitabine, a pyrimidine analog, also inhibits the growth of CC cells in a concentration-dependent manner, and the combination of gefitinib and gemcitabine showed a synergistic effect in suppressing the tumor growth of CC cells in vivo and in vitro through significant inhibition of EGFR/ERK1/2 activation relative to treatment with either gefitinib or gemcitabine alone [137]. Gemcitabine also inhibited the growth of CC cells by inducing ubiquitination-mediated degradation of EGFR and reducting of VEGF2 [138].

Recently, gefitinib, in combination with conventional chemotherapies, including gemcitabine-oxaliplatin (GEMOX) (ClinicalTrials.gov Identifier: NCT02836847) or forfirinox (ClinicalTrials.gov Identifier: NCT03768375), is under evaluation in phase 2 clinical trials for patients with recurrent ECC and GBC.

**Cetuximab.** Cetuximab is approved as an anti-EGFR monoclonal antibody for the treatment of CRC [139]. Combination treatments with cetuximab and radiotherapy for type IV CC with spine metastases showed a dramatically reduced metastatic tumor in the spine [140]. Combination treatment with cetuximab and GEMOX for patients with recurrent CC with resistance to GEMOX and high EGFR expression exhibited good efficacy and safety. The clinical outcome of cetuximab-GEMOX (gemcitabine and oxaliplatin) was one (11.1%) complete response (CR), one (11.1%) partial response (PR), one (11.1%) stable disease (SD), and six (66.7%) disease progression of nine evaluable patients. The OS was 7 months [117]. Also, the combination of GEMOX with cetuximab increases PFS (7.3 vs. 4.9 months) and OS (14.1 vs. 9.6 months) of patients with ICC relative to GEMOX alone [118]. Moreover, the treatment of cetuximab for patients with CC and GBC showed one (20%) CR, three (60%) PR, and one (20%) SD of five evaluable patients, indicating that cetuximab may be effective chemotherapy for BTC [119]. Cetuximab combined with target agents for the treatment of CC, and solid tumors are currently under evaluation in phase 1 or 2 clinical trials (ClinicalTrials.gov Identifier: NCT03829436, NCT03768375, NCT02836847, and NCT03693807).

**Panitumumab.** Panitumumab was developed as another monoclonal antibody of EGFR and is FDA approved for the first-line treatment of patients with metastatic CRC [141]. Treatment with panitumumab as monotherapy for *KRAS* wild-type irresectable BTC showed that 74% (31/42) of patients had PFS for 6 months; 3% (1/42), 31% (13/42), and 52.4% of patients showed CR, PR, and SD, respectively, and panitumumab showed reasonable efficacy (ClinicalTrials.gov Identifier: NCT00779454) [120]. Combination treatment of GEMOX with panitumumab for BTC patients, including CC, increased PFS relative to GEMOX alone (5.3 months vs. 4.4 months). GEMOX with panitumumab for patients with ICC revealed clinical benefits with improved PFS relative to that of GEMOX alone (15.1 months vs. 11.8 months) (ClinicalTrials.gov NCT01389414) [121].

Phase 2 Combination of Panitumumab and gemcitabine or Irinotecan has been evaluated in patients with metastatic BTC; 6% (2/35) and 26% (9/35) of patients had a CR and PR, respectively, and 42% (15/35) of patients showed SD. Also, 31% and 74% of patients showed the overall response rate (ORR) and disease control rate (CR + PR + SD), respectively. The one-year PFS was 44%, and one-year OS was 59%, demonstrating that this combination therapy has a clinical benefit for the treatment of BTC patients (ClinicalTrials.gov Identifier: NCT00948935) [122]. Furthermore, the combination of panitumumab with conventional agents is now under evaluation in phase 2 clinical trials in patients with CRC and CC (ClinicalTrials.gov Identifier: NCT03693807).

**Vandetanib.** The FDA-approved orphan drug vandetanib is an inhibitor of RET, VEGFR2, and EGFR for the treatment of metastatic medullary thyroid cancer. Treatment with vandetanib showed clinical benefits in CC cells. CC cells harboring the KRAS mutation and CC cells that showed the highest expression of both EGFR and VEGF were sensitive to vandetanib treatment, which suppressed EGFR activation. Vandetanib treatment also markedly inhibited tumor formation of CC cells and prolonged the time to metastasis in vivo [123]. However, the clinical trial (ClinicalTrials.gov Identifier: NCT00753675) showed that vandetanib with or without gemcitabine treatment had no effect on treatment for patients with advanced BTC and that vandetanib did not improve PFS of these patients.

**Erlotinib.** FDA-approved erlotinib is an EFGR inhibitor for first-line treatment of NSCLC patients with EGFR mutations. Administration of erlotinib exhibited clinical benefits in phase 2 clinical trials of patients with BTC. Erlotinib treatment had 8% (3/36) PR, and 17% (7/36) of patients with BTC treated with erlotinib showed no progression for six months, which was improved PFS [124]. GEMOX treatment alone and GEMOX with erlotinib for patients with BTC, including CC, were tested in phase 3 clinical trials (ClinicalTrials.gov Identifier: NCT01149122). GEMOX with erlotinib increases the PFS (5.8 months *vs.* 4.2 months) and ORR (40 *vs.* 21 patients) relative to that of GEMOX alone. Also, in CC patients, GEMOX treatment with erlotinib showed significantly increased ORR than that of GEMOX alone (30 *vs.* 12 patients) [125].

The treatment of erlotinib for a patient with hepatocholangiocellular carcinoma with the EGFR R521K mutation had an SD with no metastases and showed a response duration of more than one year [142]. Erlotinib, combined with bevacizumab as a VEGFR inhibitor, showed improved clinical outcomes: 51% (25/49) and 30.6% (15/49) of patients with CC had SD and PD, respectively (ClinicalTrials.gov Identifier: NCT00356889) [126].

**Trastuzumab and Pertuzumab.** Trastuzumab and pertuzumab are monoclonal antibodies targeting ERBB2. Recently, FDA approved pertuzumab in combination with trastuzumab, for patients with ERBB2-positive breast cancer with high-risk recurrence, and pertuzumab treatment alone showed significant improvement of invasive disease-free survival in breast cancer patients [143]. Trastuzumab as a monotherapy or in combination with lapatinib and pertuzumab for nine metastatic gallbladder cancer patients yielded 11.1% (1/9) CR, 44.5% (4/9) PR, and 33.3% (3/9) SD, but trastuzumab as monotherapy had no responses in patients with CC [144]. However, in another study, trastuzumab combined with paclitaxel or trastuzumab alone showed a dramatic regression of lung and liver metastasis of a patient with CC [127,128,129]. Also, the combination of trastuzumab and lapatinib for the treatment of CC had 75% (3/4) SD and 25% (1/4) PD [130].

Trastuzumab emtansine (T-DM1) is an antibody-drug conjugate, which is trastuzumab linked to the DM1 and was approved by the FDA for ERBB2-positive breast cancer patients with residual invasion. T-DM1 treatment significantly increased invasive disease-free survival and showed a 50% lower risk of recurrence relative to the trastuzumab group (ClinicalTrials.gov Identifier: NCT01772472) [145]. T-DM1 showed preclinical activity for inhibiting the progression of BTC, including CC. The growth inhibitory activity of BTC cells in vivo and in vitro by T-DM1 treatment was closely dependent on ERBB2 expression so that T-DM1 treatment has significant antitumor efficacy for ERBB2-expressing BTC cells but not for ERBB2-negative BTC cells in BTC xenograft models. Also, T-DM1 induces cell cycle arrest at the G2/M phase by inhibiting the activation of both ERBB2 and EGBB3 [146].

Pertuzumab, in combination with trastuzumab for ERBB2 positive one CC patient, showed an excellent, ongoing, durable response by reduction of dominant TP53 mutation and significant tumor regression [147]. However, pertuzumab, in combination with trastuzumab, has not yet been evaluated for patients with CC.

**Lapatinib.** Lapatinib, a dual inhibitor of EGFR and ERBB2, markedly reduced 80–95% of the growth of CC cells by EGFR/ERBB2 and its downstream signaling pathways, including the AKT and MAPK signaling pathways. Lapatinib treatment reduced approximately 70% of tumor formation of CC cells in vivo [41]. Lapatinib for the treatment of patients with liver cancer or BTC showed practical clinical benefits; 40% (10/25) of patients showed prolonged disease stabilization, including 60% (6/10) of patients with SD ≥ 120 days and 20% (2/10) with SD of ≥ 1 year (ClinicalTrials.gov Identifier: NCT00107536) [131]. Lapatinib treatment markedly inhibits the growth of CC cells by suppressing EGFR, human epidermal growth factor receptor (HER)2, and AKT activation relative to trastuzumab as an ERBB2 inhibitor [148]. Clinical trials with combination treatments of lapatinib and conventional chemotherapies are now under evaluation for patients with advanced or recurrent ECC and GBC (ClinicalTrials.gov Identifier: NCT03768375, NCT02836847).

**Afatinib.** Afatinib is an inhibitor for EGFR and ERBBs (HER2, HER4) and was approved by the FDA for metastatic NSCLC with non-resistant EGFR mutations (EGFR S768I, L861Q, and G719X) [149]. Afatinib, combined with paclitaxel, has been evaluated for 16 patients with solid tumors, including CC. Of these, 31% (5/16) patients with NSCLC, esophageal cancer (EC), and CC showed PR, and 50% of patients have a response duration of ≥ 6 months (ClinicalTrials.gov Identifier: NCT00809133) [150]. Afatinib markedly suppresses the viability of ICC cells by inhibiting EGFR/STAT3 activation [132]. Interestingly, afatinib treatment revealed a significant, durable response to CC and NSCLC patients with NRG1 fusion proteins such as ATP1B1-NRK1 and SDC4-NRG1 [151]. Treatments of afatinib are also now under evaluation for patients with BTC (ClinicalTrials.gov Identifier: NCT02451553, NCT02465060).

**Chimeric antigen receptor (CAR)-T-cell therapy.** The FDA approved the chimeric anti-CD19 antigen receptor (CAR)-T-cell immunotherapies (tisagenlecleucel and axicabtagene clioleucel) for patients with B-cell lymphoma and acute lymphoblastic leukemia. Various EGFR-and ERBB2-CAR-T clinical trials are also under evaluation for patients with various cancer types (ClinicalTrials.gov Identifier: NCT03618381, NCT03638167, NCT03696030, NCT03198052, etc.). EGFR-CAR-T and ERBB2-CAR-T immunotherapy have been evaluated in NSCLC and sarcoma patients, respectively, and are safe and feasible for treatment. The EGFR-CAR-T cell infusions for NSCLC patients showed 18.2% (2/11) OR and 45.5% (5/11) SD for two to eight months (NCT01869166) [152]. Additionally, the ERBB2-CAR-T cell infusions for NSCLC patients showed 23.5% (4/17) SD for 12 weeks to 14 months; 17.7% (3/17) patients underwent surgical resection of their residual tumors, and one of these patients showed ≥ 90% tumor necrosis (ClinicalTrials.gov identifier: NCT00902044) [153].

Successive infusions of EGFR-CAR-T and CD133-CAR-T immunotherapy showed clinical benefits for the treatment of CC: EGFR-CAR-T therapy had PR for 8.5 months, and CD133-CAR-T therapy achieved PR for 4.5 months (ClinicalTrials.gov Identifier: NCT01869166 and NCT02541370) [133]. Another study demonstrated that EGFR-CAR-T cell infusions in phase 1 clinical trial for patients with BTC, including CC, have a safe and improved clinical outcome (ClinicalTrials.gov identifier: NCT01869166); 6% (1/17) and 58.8% (10/17) of these patients had CR for 22 months and SD for 2.5 to 15 months, respectively [134]. Treatment of MUC-1-CAR-T immunotherapy is now under evaluation for patients with CC (ClinicalTrials.gov Identifier: NCT03633773).

**Nimotuzumab.** Nimotuzumab is a monoclonal antibody of human EGFR that has been approved in several countries for glioma, head and neck cancer, and esophageal cancer. Nimotuzumab alone and with conventional agents are now under evaluation in various clinical trials, and it has shown potential efficacy in various clinical trials, including nasopharyngeal carcinoma [154], esophageal cancer [155], and glioma [156].

Nimotuzumab treatment markedly reduces the growth and metastatic ability of CC cells by suppressing EGFR/AKT/p38 signaling, MMP9 expression, and inhibiting EMT [135]. However, nimotuzumab alone or with other conventional agents has still not yet been evaluated in clinical trials for patients with CC.

**Tucatinib.** Tucatunib was developed as an ERBB2 inhibitor, and the FDA approved its use in combination with trastuzumab and capecitabine for ERBB2-positive metastatic breast cancer patients in April 2020. The PFS (33% vs. 12.3%) and duration of PFS (7.8 months vs. 5.6 months) at one year in patients with a combination of tucatinib plus trastuzumab and capecitabine were markedly increased relative to those of the trastuzumab- and capecitabine-treated groups, respectively. Additionally, the risk of disease progression or death was markedly decreased by 46% in patients with a combination of tucatinib plus trastuzumab and capecitabine compared with that of patients receiving trastuzumab and capecitabine. Moreover, the OS of the tucatinib plus trastuzumab and capecitabine group at 2 years significantly increased over that of the group without tucatinib (44.9% vs. 26.6%). In the cases of patients with brain metastases, PFS at 1-year follow-up in patients receiving tucatinib plus trastuzumab and capecitabine was drastically increased relative to that of patients with trastuzumab and capecitabine (24.9% vs. 0%) (ClinicalTrials.gov Identifier: NCT02614794) [157,158]. These results indicated that tucatinib with trastuzumab and capecitabine showed better clinical benefits than adding a placebo. Recently, tucatinib has been under evaluation for patients with breast cancer and CRC in combination with trastuzumab and capecitabine or palbociclib and letrozole (ClinicalTrials.gov Identifier: NCT03054363, NCT03501979, NCT03043313). Nevertheless, tucatinib, in combination with trastuzumab and capecitabine, has still not been applied in clinical trials for patients with BTC, including CC.

## 8. Conclusions

Members of the ERBB family, such as EGFR and ERBBs, are one of the best-studied receptor tyrosine kinases during the progression of various types of cancer. Proliferation, tumorigenesis, metastasis, and chemoresistance of various types of cancer, including CC, are induced by activation of the EGBB family. However, the precise role of EGFR and ERBBs alone or in crosstalk with other oncoproteins or transmembrane receptors in the development and progression of CC has remained largely unknown and unclear compared with those of other cancer types. It will be important to investigate how ERBB family members contribute to CC cancer progression in more detail to identify unpredicted functions of this protein family and to develop potential therapeutic strategies with other conventional therapeutic agents for expanding clinical trials. Recently, mutations of EGFR and ERBBs were identified, but the functional role of the newly identified mutations of EGFR and ERBBs has remained unclear. However, these mutations seem to affect the survival, invasion, metastasis, and chemoresistance of CC through changes in the expression or activation of EGFR and ERBBs. Overexpression and activation by mutation of EGFR and ERBBs have been linked to the development and progression of CC and acquisition of chemoresistance against chemotherapeutic agents. These have led to the new development of therapeutic agents and strategies for targeting EGFR and ERBBs, including new combination strategies with other tyrosine kinase inhibitors or conventional inhibitors to repress the activation or expression of EGFR or ERBBs. The development of such new strategies must be founded on the identification of new mechanisms and biomarkers by unraveling the functional roles of EGFR or ERBBs in the progression of CC.

## Figures and Tables

**Figure 1 jcm-09-02255-f001:**
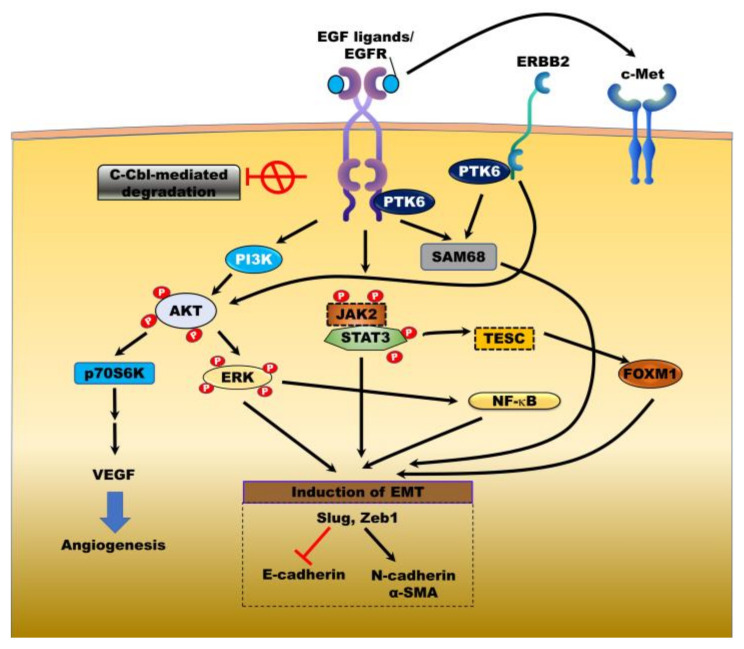
Mechanisms of the EGFR family signaling in cholangiocarcinoma. ErBb family leads to tumor growth, vascularization, tumorigenesis, and metastasis through ligand-mediated activation of downstream targets and other transmembrane receptors, c-Met. Inhibition of c-cbl-mediated degradation of epidermal growth factor receptor (EGFR) enhances accumulation and activation of EGFR and then activates the downstream signaling pathways and contribute to angiogenesis and epithelial-mesenchymal transmission (EMT) program by induction of vascular endothelial growth factor (VEGF) expression and EMT transcription factors (EMT-TF’s). JAK2: Janus kinase 2, STAT3: Signal transducer and activator of transcription 3, SAM68: KH RNA binding domain containing, signal transduction associated 1, PI3K: Phosphatidylinositol-4,5-bisphosphate 3-kinase, ERK: Mitogen-activated protein kinase, VEGF: Vascular endothelial growth factor, NF-κB: Nuclear factor kappa B, FOXM1: Forkhead box M1, TESC: Tescalcin, Slug: Snail family transcriptional repressor 2, Zeb1: Zinc finger E-box binding homeobox 1, PTK6: Protein tyrosine kinase 6.

**Figure 2 jcm-09-02255-f002:**
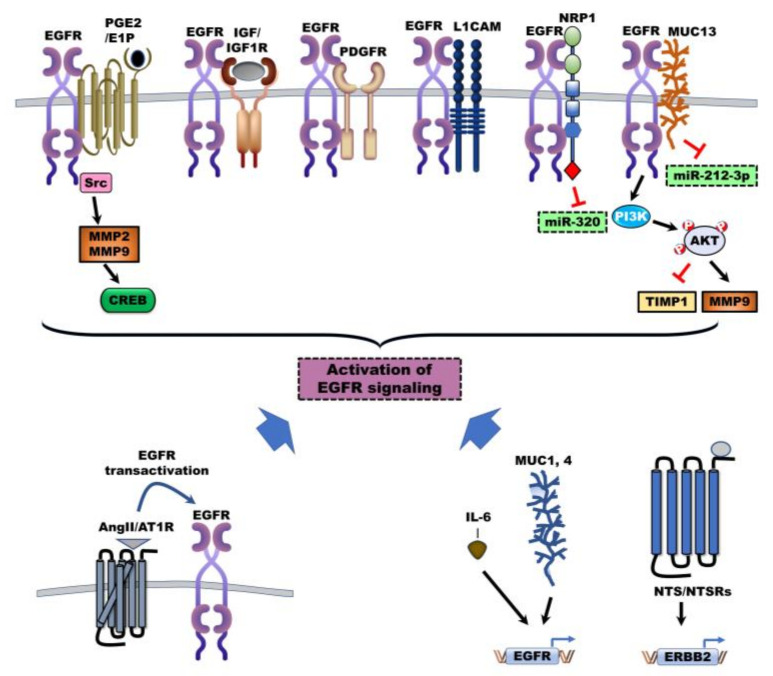
Mechanisms of other ligands and its receptor as positive modulators of (ErBb) family signaling activation in cholangiocarcinoma. Other ligands and receptors as positive modulators contribute to the activation of the ErBb family via several mechanisms to activate ErBbs signaling in the progression of cholangiocarcinoma (CC). Interleukin (IL)-6, mucins, and neurotensin (NTS)/NTS receptors-mediated EGFR and ERBB2 overexpression leading to activation of (ErBb) family signaling. EGFR also can be activated by interaction with other receptors and angiotensin II (Ang II) mediated activation of G protein-coupled receptor (GPCR) lead to the transactivation of EGFR via cross-talk between GPCR and EGFR. PGE2: Prostaglandin, IGF: Insulin-like growth factor, IGF1R: Insulin-like growth factor-1 receptor, MMPs: Matrix metallopeptidases, CREB: Cyclic AMP response element-binding protein, PDGFR: Platelet-derived growth factor, L1CAM: L1 cell adhesion molecule, NRP1: Neuropilin-1, MUC: mucin, TIMP1: TIMP metallopeptidase inhibitor 1, Ang II: Angiotensin II, AT1R: Angiotensin receptor, NTS: Neurotensin, NTSR: NTS receptor.

**Figure 3 jcm-09-02255-f003:**
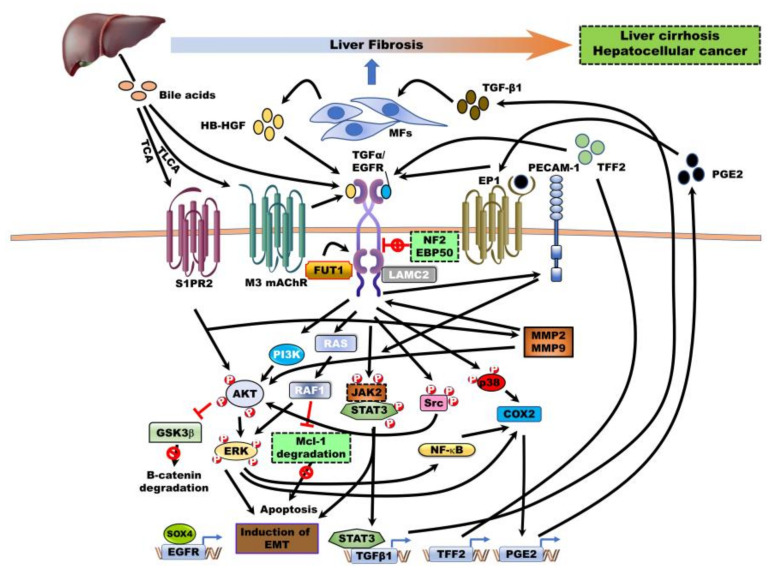
The contributions of the tumor microenvironments and other regulators to the activation of EGFR in cholangiocarcinoma. Secretion of heparin-binding EGF-like growth factor (HB-EGF) by human liver myofibroblasts (HLMF’s) and elevated levels of bile acids from the liver can be overstimulating EGFR in paracrine fashions. EGFR can also be activated by secretion of transforming growth factor (TGF)-β1, the trefoil factor family (TFF) 2, prostaglandin (PGE) 2 in an autocrine fashion, which is produced by the CC cells. Moreover, other positive or negative regulators are involved in the regulation of EGFR activation by direct interaction. TGFα: Transforming growth factor-α, HB-EGF: EGF-like growth factor, TLCA: Taurolithocholic acid, TCA: Taurocholic acid, S1PR2: Sphingosine 1-phosphate receptor 2, M3 mAChR: Muscarinic acetylcholine receptor M3, FUT1: fucosyltransferase 1, LAMC2: Laminin subunit gamma 2, NF2: Neurofibromatosis type 2, EBP50: SLC9A3 regulator 1, EP1: Prostaglandin E receptor 1, TFF2: Trefoil factor family 2, GSK3β: Glycogen synthase kinase 3, MCL1: MCL1 apoptosis regulator, SOX4: SRY-box transcription factor 4, COX2: Cyclooxygenase 2, TGF-β1: Transforming growth factor- β1.

**Figure 4 jcm-09-02255-f004:**
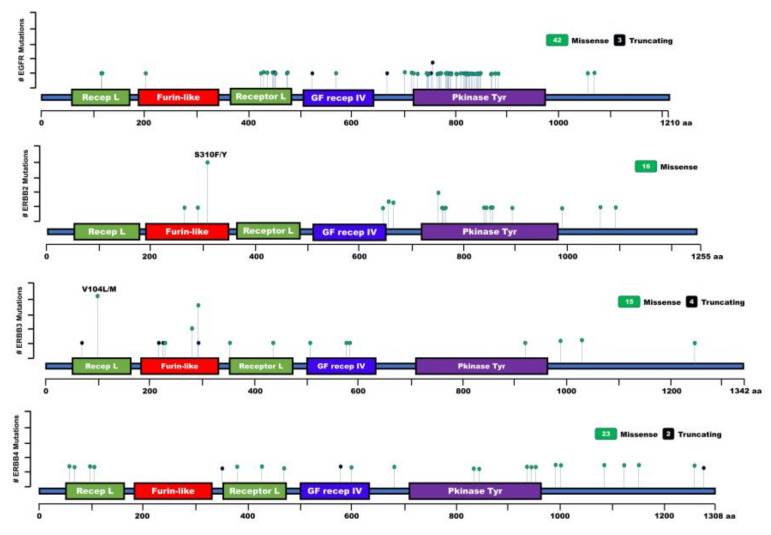
Lollipop plot showing the distribution and classes of mutations in EGFR, ERBB2, ERBB3, and ERBB4, discovered in the cholangiocarcinoma. Green circles indicate missense mutations, and black circles indicate truncating mutations. Recep L: Receptor L domain, Furin-like: furin-like cysteine-rich region, GF Recep IV: growth factor receptor domain IV, Pkinase Tyr: protein tyrosine kinase domain. aa: Amino acid.

**Table 1 jcm-09-02255-t001:** Identification of Mutations of Epidermal growth factor receptor (EGFR) Families in Cholangiocarcinoma.

Protein	Cancer Type	Protein Change	Clinical Implications	Mutation Type	Refs
EGFR	ICC	K745_E749 del	Oncogenic	In-F del	[102]
	ECC	E746_A750 del	Oncogenic	In-F del	[102]
	ICC	K575R	Oncogenic	Missense	[103]
	ICC	E872K	Oncogenic	Missense	[103]
	ICC	T790M	Oncogenic	Missense	[103]
	ECC	C775Y	Oncogenic	Missense	[103]
	ECC	G882S	Oncogenic	Missense	[103]
	ECC	V843I	Oncogenic	Missense	[103]
	ECC	L858	Oncogenic	Missense	[103]
	GBC	A864T	Oncogenic	Missense	[103]
	CC	E804K	Oncogenic	Missense	[104]
	ICC	G719S	Oncogenic	Missense	[105]
	ICC	E709K	Oncogenic	Missense	[106]
	ICC	L747_P753 del S ins	Unknown	In-F del	[106]
	ICC	V786M	Oncogenic	Missense	[106]
	CC	G719X	Unknown	Missense	[107]
	CC	S768I	Oncogenic	Missense	[107]
	CC	L861Q	Oncogenic	Missense	[107]
	ICC	T783I	Oncogenic	Missense	[108]
	ICC	S784F	Oncogenic	Missense	[108]
	ICC, ECC	D837N	Oncogenic	Missense	[108]
	ECC	D800G	Oncogenic	Missense	[108]
	ECC	C818R	Oncogenic	Missense	[108]
	ECC	V819M	Oncogenic	Missense	[108]
	ECC	Q820R	Oncogenic	Missense	[108]
	ECC	V851I	Oncogenic	Missense	[108]
	ECC	G873E	Oncogenic	Missense	[108]
	ECC	G874D	Oncogenic	Missense	[108]
	BTC	L443Q	Unknown	Missense	[109]
	BTC	S464P	Oncogenic	Missense	[109]
	BTC	K467*	Unknown	Missense	[109]
	BTC	N468D	Unknown	Missense	[109]
	BTC	G482E	Unknown	Missense	[109]
	BTC	G482R	Unknown	Missense	[109]
	BTC	L469S	Unknown	Missense	[109]
	BTC	L707S	Oncogenic	Missense	[109]
	ECC	E114K	Oncogenic	Missense	[110,111]
	ICC	Y1069C	Oncogenic	Missense	[110,111]
	ICC	I425L	Oncogenic	Missense	[110,111]
	ICC	C818F	Likely oncogenic	Missense	[110,111]
	ICC	R669Q	Oncogenic	Missense	[110,111]
	ICC	V524S	Likely oncogenic	Missense	[110,111]
	GBC	G203R	Oncogenic	Missense	[110,111]
	ICC	R427H	Oncogenic	Missense	[112]
	ICC	R324L	Oncogenic	Missense	[112]
ERBB2	ICC	L755P/S	Oncogenic	Missense	[111]
	ICC	S310F	Oncogenic	Missense	[111,113]
	ICC	L994V	Unknown	Missense	[111]
	GBC, ICC	V842I	Oncogenic	Missense	[111,112]
	GBC, ECC	S310Y	Oncogenic	Missense	[111,112]
	GBC, ECC	G292R	Oncogenic	Missense	[111,112]
	GBC	E265K	Oncogenic	Missense	[111]
	GBC	L1098M	Unknown	Missense	[111]
	ICC	L869R	Oncogenic	Missense	[112]
	ECC	G660D	Oncogenic	Missense	[112]
	ICC	R897W	Oncogenic	Missense	[112]
	ECC	D769H	Oncogenic	Missense	[112]
	ICC, ECC	R678Q	Oncogenic	Missense	[112]
	ECC	T862A	Oncogenic	Missense	[112]
	ICC	S653C	Oncogenic	Missense	[112]
	ECC	G776V	Oncogenic	Missense	[112]
ERBB3	ECC	V104M	Oncogenic	Missense	[110,111]
	ECC	A232V	Oncogenic	Missense	[110,111]
	ECC	G582W	Oncogenic	Missense	[110,111]
	GBC, ICC	G284R	Oncogenic	Missense	[110,111]
	GBC	V104L	Oncogenic	Missense	[110,111]
	GBC, ICC	D297Y	Oncogenic	Missense	[110,111]
	GBC	T355I	Oncogenic	Missense	[110,111]
	GBC	V1035D	Unknown	Missense	[110,111]
	GBC	R444Q	Oncogenic	Missense	[110,111]
	GBC	V586M	Unknown	Missense	[110,111]
	GBC	G994D	Unknown	Missense	[110,111]
	ICC	E928G	Oncogenic	Missense	[110,111]
	ICC	G508R	Oncogenic	Missense	[110,111]
	ICC	A1252S	Unknown	Missense	[110,111]
	ICC	D581N	Unknown	Missense	[110,111]
	ICC	N222Tfs*47	Unknown	FS del	[110,111]
	ICC	E230Dfs*39	Unknown	FS del	[110,111]
	ICC	D73Tfs*11	Unknown	FS del	[110,111]
	ICC	D297Ifs*16	Unknown	FS del	[110,111]
ERBB4	ICC	R103C	Oncogenic	Missense	[110,111]
	ICC	I68N	Oncogenic	Missense	[110,111]
	ICC	L432M	Oncogenic	Missense	[110,111]
	ICC	S602C	Unknown	Missense	[110,111]
	ICC	P1158H	Unknown	Missense	[110,111]
	ICC	S602C	Unknown	Missense	[110,111]
	ICC	S1286Lfs*5	Unknown	FS del	[110,111]
	ICC	F356Sfs*2	Unknown	FS del	[110,111]
	GBC	T475A	Unknown	Missense	[110,111]
	ECC	Q1126K	Unknown	Missense	[112]
	ECC	E835D	Unknown	Missense	[112]
	ECC	F689L	Unknown	Missense	[112]
	ECC	K682N	Oncogenic	Missense	[112]
	ECC	P1092S	Unknown	Missense	[112]
	ECC	R106C	Oncogenic	Missense	[112]
	ECC	S522L	Oncogenic	Missense	[112]
	ECC	S79Y	Oncogenic	Missense	[112]
	ICC	C580*	Oncogenic	Nonsense	[112]
	ICC	D376Y	Unknown	Missense	[112]
	ICC	D960G	Oncogenic	Missense	[112]
	ICC	Q1270K	Oncogenic	Missense	[112]
	ICC	R847C	Oncogenic	Missense	[112]
	ICC	R992H	Oncogenic	Missense	[112]
	ICC	Y1066H	Unknown	Missense	[112]
	ICC	Y950H	Unknown	Missense	[112]

CC: Cholangiocarcinoma; BTC: Biliary Tract Cancer; ECC: Extrahepatic Cholangiocarcinoma; ICC: Intrahepatic Cholangiocarcinoma; GBC: Gallbladder Cancer; In-F del: In-Frame Deletion; FS: Del: Frameshift Deletion.

**Table 2 jcm-09-02255-t002:** ERBB Family Protein inhibitors in Combination with Other Inhibitors and Their Efficacy in Treating Cholangiocarcinoma.

**Inhibitor/Clinical Trial Identifier**	**Phase**	**Efficacy of Drug**	**Refs**
Imatinib + Fluorouracil (5-FU) or Leucovorin /NCT01153750	Cell line/ Phase 2	Imatinib inhibits platelet-derived growth factor receptor β (PDGFRβ)-mediated cell proliferation and tumor growth/under evaluation in clinical trials.	[114,115]
Gefitinib + gemcitabine–oxaliplatin (GEMOX) or forfirinox/NCT02836847, NCT03768375	Cell line/ Phase 2	Gefitinib + CI-1040 significantly suppressed ~60% of tumor growth of CC cells, and gefitinib + gemcitabine showed a synergistic effect in suppressing the tumor growth of CC cells/under evaluation for patients with recurrent ECC and GBC.	[116]
Cetuximab + GEMOX/NCT03829436, NCT03768375, NCT02836847, NCT03693807	Phase 1/2	Cetuximab + radiotherapy for type IV CC with spine metastases showed a dramatically reduced metastatic tumor in the spine. Also, cetuximab + GEMOX showed good efficacy and safety of cetuximab/Cetuximab combined with target agents, is under evaluation in clinical trials.	[117,118,119]
Panitumumab + gemcitabine or Irinotecan or conventional agents/NCT01389414, NCT03693807	Phase 2/3	3%, 31%, and 52.4% of patients with treatment of panitumumab showed CR, PR, and SD, respectively, and panitumumab showed reasonable efficacy. Also, GEMOX + panitumumab revealed clinical benefits with improved PFS. Also, Panitumumab + gemcitabine or Irinotecan therapy has a clinical benefit for the treatment of BTC patients/under evaluation for patients with CRC and CC.	[120,121,122]
Vandetanib/NCT00753675	Cell line/ Phase 2	vandetanib markedly inhibited tumor formation of CC cells, but vandetanib + gemcitabine treatment did not affect patients with advanced BTC.	[123]
Erlotinib + or GEMOX or bevacizumab/NCT01149122, NCT00356889	Phase 2/3	This treatment has 8% PR, and 17% of patients with BTC treated with erlotinib showed no progression for 6 months. GEMOX with erlotinib increases the PFS. Also, Erlotinib + bevacizumab showed 51% (25/49), and 30.6% (15/49) of patients with CC had SD and PD, respectively	[124,125,126]
Trastuzumab + lapatinib or pertuzumab or tucatinib/NCT01772472, NCT04430738 NCT03613168	Phase 1/2/3	Trastuzumab, combined with paclitaxel showed a dramatic regression of lung and liver metastasis of a patient with CC. Also, trastuzumab + lapatinib for the treatment of CC had 75% SD and 25% PD. Moreover, T-DM1 treatment significantly increased invasive disease-free survival	[127,128,129,130]
Lapatinib + GEMOX or conventional agents/NCT00107536, NCT00350753, NCT03768375, NCT02836847	Phase 2	40% of patients with lapatinib showed prolonged disease stabilization. Lapatinib + conventional chemotherapies are now under evaluation in clinical trials.	[131]
Afatinib/NCT02451553, NCT02465060	Cell line/ Phase 1/2	Afatinib markedly suppresses the viability of ICC cells by inhibiting EGFR/STAT3 activation/now under evaluation in clinical trials for patients with BTC.	[132]
CAR-T immunotherapy/NCT01869166, NCT01869166 NCT02541370	Phase 1/2	EGFR-CAR-T and CD133-CAR-T immunotherapy have a safe and improved clinical outcome: EGFR-CAR-T therapy had PR for 8.5 months, and CD133-CAR-T therapy achieved PR for 4.5 months. MUC-1-CAR-T immunotherapy is now under evaluation for patients with CC.	[133,134]
Nimotuzumab	Cell line	Nimotuzumab markedly reduces the growth and metastatic ability of CC cells	[135]

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
