# Peer review of "ErBb Family Proteins in Cholangiocarcinoma and Clinical Implications"

_jcm, 2020, doi:10.3390/jcm9072255_

Round 1
Reviewer 1 Report
Dear Author,
Thank you very much for submitting your work to Journal of Clinical Medicine.
This is a very interesting narative review regarding the role of ErBb proteins in the development and treatment of cholangiocarcinoma.
The effort is commendable as the article is remarkably comprehensive and well structured. All necessary information are provided for the readers to understand current knowledge on molecular oncology. Despite the extent of information, the manuscript is well organized and easy for the reader to follow.
All relevant therapeutic implementations have been extensively discussed within the context of definite evidence for other types of malignancies as well as ongoing research in cholangiocarcinoma.
If I had to suggest for a few minor changes, these would be:
a) avoiding personal pronouns (such as "we", "our", etc.) throughout the manuscript.
b) including a table with discussed treatments along with most advanced trial phase.
Overall this is an excellent articleand I would be more than happy to review a revised version.
Kind regards
Author Response
If I had to suggest for a few minor changes, these would be:
a) avoiding personal pronouns (such as "we", "our", etc.) throughout the manuscript.
As your suggestion, we removed personal pronouns throughout the manuscript.
b) including a table with discussed treatments along with most advanced trial phase.
As your suggestion, we made a new table with the most advanced trial phase. Please see Table 2.
Overall this is an excellent article and I would be more than happy to review a revised version.
Thank you for your support and your suggestion.
Reviewer 2 Report
The manuscript gives a comprehensive and detailed overview of ErBb family of proteins, by dissecting their mechanisms of action, mutational landscape, and most importantly, the authors also provide a therapeutic targeting of ErBb family in cholangiocarcinoma. The review is well structured and clearly written.
If authors agree, I would only suggest to increase the font size in figures, for better visibility.
I congratulate the authors on their excellent work!
Best regards!
Author Response
If authors agree, I would only suggest to increase the font size in figures, for better visibility.
Thank you for your support and your suggestion. As your suggestion, we have increased the font of the picture as much as possible.
Reviewer 3 Report
Wook Jin provides a comprehensive review on cholangiocarcinoma with a focus on ErBb family proteins in the disease process. The review is carefully written and well structured and moreover summerizes current knowledge on CCA.
I do not have major concerns but a few minor points.
1) Most of the review is focused on the role of EGFR (ERBB1) in cholangiocarcinoma, how about mutations and inhibitors for other ErBb members?
2) In terms of the therapeutic options, the manuscript lacks a detailed table or figure to explicitly enumerate all current or pending options under development. I believe the addition of this aspect will offer a better direction for other scientists in this field.
3) Also, some grammatical issue should be addressed
Author Response
1) Most of the review is focused on the role of EGFR (ERBB1) in cholangiocarcinoma, how about mutations and inhibitors for other ErBb members?
Thank you for your comments. We listed mutations for ERBB2, ERBB3, and ERBB4 from line 386 to 400 in the review manuscript. However, we have tried, but the role and function of mutations in ERBB2, 3, and 4 related to cholangiocarcinoma can no longer be found in previous studies. Sorry about that, and please understand this.
In the case of inhibitors for other ERBB members, we summarized trastuzumab, lapatinib, tucatinib as an ERBB2 inhibitor in our review manuscript. We also summarized Afatinib as ERGF and ERBBs (ERBB2 and ERBB4) inhibitor and ERBB2-CAR-T immunotherapy in our review manuscript.
2) In terms of the therapeutic options, the manuscript lacks a detailed table or figure to explicitly enumerate all current or pending options under development. I believe the addition of this aspect will offer a better direction for other scientists in this field.
Thank you for your recommendation. As your suggestion, we summarized table 2 with discussed treatments along with the current or pending trial phase for cholangiocarcinoma. Please see Table 2.
3) Also, some grammatical issue should be addressed
We apologize for this error and have corrected this through checking by proofreading this manuscript in English proofreading & editing service with native English speakers to correct grammatical errors and to improve our manuscript.
Reviewer 4 Report
The authors in the work have extensively described the ErBb signaling in cholangiocarcinoma and have also described the current therapy in trials. The description is sufficient and extensive.
Author Response
The authors in the work have extensively described the ErBb signaling in cholangiocarcinoma and have also described the current therapy in trials. The description is sufficient and extensive.
Thank you for your support.
Round 2
Reviewer 1 Report
Dear Authors,
Thank you for responding to my comments.
I find your response adequate and am happy to recommend in favor of publication of the manuscript.